# HBV cccDNA: The Molecular Reservoir of Hepatitis B Persistence and Challenges to Achieve Viral Eradication

**DOI:** 10.3390/biom15010062

**Published:** 2025-01-04

**Authors:** André Boonstra, Gulce Sari

**Affiliations:** Department of Gastroenterology and Hepatology, Erasmus Medical Center, Wytemaweg 80, 3015CN Rotterdam, The Netherlands

**Keywords:** hepatitis B, covalently closed circular (ccc) DNA, viral reservoir

## Abstract

Hepatitis B virus (HBV) is a major global health issue, with an estimated 254 million people living with chronic HBV infection worldwide as of 2022. Chronic HBV infection is the leading cause of cirrhosis and liver cancer. Current treatment with nucleos(t)ide analogs is effective in the suppression of viral activity but generally requires lifelong treatment. They fail to eradicate the HBV viral reservoir, called covalently closed circular DNA (cccDNA), which replicates in the nucleus of liver cells. The cccDNA serves as the sole template for viral replication, as it generates the pregenomic RNA (pgRNA) necessary for producing new viral genomes. This stable form of viral DNA can reactivate the virus when treatment is stopped. HBV cccDNA is therefore one of the main challenges in curing chronic HBV infections. By targeting steps such as cccDNA formation, capsid assembly, or particle secretion, researchers continue to seek ways to interfere with HBV replication and to reduce its persistence, ultimately to eradicate HBV as a global health problem. This review provides an overview of what is currently known about cccDNA formation and biogenesis and the ongoing efforts to target and eradicate it to cure chronic HBV infections.

## 1. Introduction

Hepatitis B virus (HBV) is an infectious virus that primarily targets the liver, causing both short-term (acute) and long-term (chronic) liver diseases. HBV is a major global health issue, with an estimated 254 million people living with chronic HBV infection worldwide as of 2022, according to the World Health Organization (WHO) [1]. Chronic HBV infections are the leading cause of cirrhosis and liver cancer, contributing to a large number of liver-related deaths worldwide [1].

The most common transmission route for HBV is from mother to child at birth (vertical transmission). However, transmission can also occur through contact with infected bodily fluids through shared syringes, contaminated medical equipment, and sexual contact with an infected person. If the infection is symptomatic in the acute phase, then jaundice, fatigue, abdominal pain, nausea, and dark urine can be observed. In contrast to adults who generally resolve the infection spontaneously and fully recover from the primary infection, most infected neonates fail to clear the infection and develop a chronic infection, which can lead to progressive liver damage, cirrhosis, and a heightened risk of liver cancer [2,3,4,5].

A highly effective prophylactic vaccine is available, which is part of childhood immunization programs worldwide; the vaccine is safe and provides long-lasting protection. For those with chronic HBV infection, treatment options include antiviral medications, such as nucleos(t)ide analogs (NAs) tenofovir and entecavir, or pegylated interferon-α (IFNα) therapies. The main goal of treatment is to prevent the disease from progressing to cirrhosis or liver cancer. Pegylated-IFNα has the potential to cure chronic HBV infection in a small percentage of patients; however, its use is limited by significant side effects, making it less viable as a long-term treatment option [2,3,6,7]. Treatment of chronic HBV patients with NA on the other hand leads to suppression of HBV replication, but patients have to take their medication lifelong. The reason for this is that a viral reservoir, called covalently closed circular DNA (cccDNA), remains present in the nucleus of liver cells. This stable form of viral DNA can reactivate the viral replication when treatment is stopped, and this cccDNA is therefore one of the main challenges in curing chronic HBV infections. In fact, current antiviral therapies are ineffective in eliminating cccDNA, making it a key obstacle to fully eradicating HBV [6,7,8,9,10,11].

All in all, achieving viral suppression without viral eradication requires long-term treatment, possibly lifelong, and even though healthcare costs for HBV treatments vary significantly worldwide. In the United States, they can amount to USD 13,000 to USD 20,000 per patient each year [12]. Since cccDNA is central to HBV’s ability to persist and cause chronic infections, understanding how it forms, functions, and remains in the infected hepatocytes is critical for developing more effective treatments. This review provides an overview of what is currently known about cccDNA and the ongoing efforts to target and eradicate it to cure chronic HBV infections.

## 2. HBV Genome and Structure: A Journey from Infection to Circulation

HBV belongs to the *Hepadnaviridae* family, specialized in infecting hepatocytes—the primary cell type of the liver. As a partially double-stranded DNA virus, HBV’s genome is approximately 3.2 kilobases (kb) in length, with a unique structure that plays a key role in its life cycle and infectivity. This viral genome comprises a full-length negative strand, which harbors a viral polymerase enzyme attached at its 5′ end, and a shorter, incomplete positive strand. Together, these two DNA strands form what is referred to as “relaxed-circular DNA” (rcDNA), a specialized structure that is fundamental to HBV’s replication cycle [13].

Upon entry into a hepatocyte, the HBV rcDNA genome is transported to the nucleus. Within this protected environment, and with the aid of host cellular factors, the transcriptionally inert rcDNA undergoes conversion into the fully double-stranded cccDNA. This stable mini-chromosome, established in the nucleus, is critical to HBV persistence, as it serves as the template for the transcription of viral RNAs. These RNAs are divided into genomic and sub-genomic types and are essential for producing the proteins and genomic material needed to propagate new viral particles [14,15].

The HBV genome encodes four overlapping open reading frames (ORFs)—C, P, S, and X—each responsible for generating specific viral proteins that fulfill different roles in the virus’s life cycle (Figure 1). The *ORF-C* produces the core protein (HBcAg), which forms the viral capsid, as well as the E antigen (HBeAg). The *ORF-P* encodes the viral DNA polymerase, essential for replication. The *ORF-S* is unique in its versatility, processing into three different surface antigens—small, medium, and large (S, M, and L) HBsAg—all of which incorporate into the viral envelope and aid in viral entry. Finally, *ORF-X* encodes the regulatory protein X, a multifunctional protein involved in modulating host–cellular responses to benefit viral persistence and replication [14,15].

The cccDNA serves as the sole template for viral replication, as it generates the pregenomic RNA (pgRNA) necessary for producing new viral genomes. Following transcription, the pgRNA is packaged with the viral polymerase within newly formed capsids. Within these capsids, the pgRNA is reverse transcribed into the rcDNA, thus completing the formation of a new viral genome that will be ready for secretion or reinfection. The HBV capsid itself, composed of 90–120 HBcAg dimers, is a sturdy icosahedral structure that shields and transports the viral genome. Besides protecting the genome, the capsid interacts with various viral proteins, including HBeAg and HBsAg, and connects them with intracellular membranes to initiate viral secretion [16]. As the HBV life cycle reaches completion, three distinct particle types are released into the bloodstream: infectious HBV particles, non-infectious spherical particles, and filamentous particles.

Infectious HBV particles (Dane particles) are spherical, complete viral particles encapsulated within a lipid membrane, and they have incorporated HBsAg on their surface and HBeAg between the capsid and lipid membrane. Within the nucleocapsid, they carry the rcDNA genome and viral polymerase. Dane particles are the infectious form of HBV and are directly responsible for propagating infection (Figure 1A).

The non-infectious spherical particles are smaller, rounded particles and structurally resemble Dane particles, but they lack viral DNA, rendering them non-infectious (Figure 1B) but they contain HBsAg, dominantly S and M HBsAg. Lastly, the filamentous particles are elongated structures of varying lengths and diameters, but they also lack viral genomes, making them non-infectious (Figure 1C). The filamentous particles, like the smaller spheres, carry HBsAg [16]. Exact roles of non-infectious spherical and filamentous particles are not yet known. Additionally, patient serum has been found to contain other non-infectious HBV-related particles that carry viral RNAs rather than DNA [17,18].

Understanding the complex life cycle of HBV and its unique methods for interacting with host cells provides critical insights into potential therapeutic strategies. By targeting steps such as cccDNA formation, capsid assembly, or particle secretion, researchers continue to seek ways to interfere with HBV replication and reduce its persistence, ultimately aiming to advance treatments for hepatitis B and mitigate its effects on global health.

## 3. HBV Life Cycle: From Cell Entry to Viral Progeny Release

Although primarily responsible for importing bile salts into liver cells, the liver-specific sodium taurocholate co-transporting polypeptide (NTCP) receptor on hepatocytes was identified as a key entry receptor for HBV, revealing a fascinating hijack mechanism [19,20]. The virus specifically binds to NTCP via an extended region on its large HBsAg protein, known as the preS1 domain [19]. This interaction initiates viral internalization through an intricate process involving a recently identified helper receptor, the epidermal growth factor receptor (EGFR), which collaborates with NTCP to mediate endocytosis of the viral particle into the hepatocyte [19,20,21,22] (Figure 2, steps 1 and 2). 

A critical step in viral entry is the polymerization of NTCP triggered by its interaction with the HBV preS1 domain. This polymerization process is essential for viral uptake, as inhibitors targeting NTCP polymerization have been shown to effectively block HBV internalization [23]. Once inside the cell, the viral envelope fuses with the host endocytic membrane, enabling the release of the viral nucleocapsid into the cytoplasm. Guided by cellular microtubules, the nucleocapsid is transported to the nuclear membrane, where it gains access to the nucleus through the nuclear pore complex [24,25] (Figure 2 step 3). Within the nucleus, the HBV nucleocapsid undergoes disassembly, allowing for the release of the rcDNA viral genome. The viral polymerase, initially bound to the rcDNA, is removed, and a series of host cell factors mediate the repair of rcDNA gaps to form the fully double-stranded cccDNA [26,27,28]. This cccDNA minichromosome serves as the central template for all subsequent viral replication activities. Interestingly, small fragments of the HBV genome also integrate into the host’s genome, a process that does not generate new viral particles but does facilitate the production of certain viral proteins, such as HBsAg [29] (Figure 2, steps 4 and 5).

Once established, the cccDNA can be transcribed by host RNA polymerase II activity, generating HBV RNAs of varying lengths (3.5, 2.4, 2.1, and 0.7 kb) encoding for viral proteins and genomic material necessary for replication. Among these, the longest 3.5 kb RNA transcript contains the preC RNA and pgRNA, which encode HBcAg, HBeAg, and viral polymerase. The 2.4 kb RNA is dedicated to the production of large HBsAg, while the 2.1 kb RNA encodes the medium and small HBsAg isoforms. Lastly, the smallest 0.7 kb RNA transcript mediates the generation of the regulatory HBx protein, which plays a key role in modulating host cell functions (reviewed in detail in [30]). Each HBV RNA transcript possesses a polyadenine (PolyA) tail, crucial for the RNA’s stability and its export out of the nucleus into the cytoplasm for translation.

Within the cytoplasm, translation of these RNAs occurs primarily at the endoplasmic reticulum (ER), where HBV proteins are synthesized. HBcAg proteins self-associate to form the icosahedral capsid, encapsulating the pgRNA and viral polymerase. Within this capsid, the viral replication process begins with the reverse transcription of pgRNA into a complementary negative DNA strand. As this synthesis progresses, the pgRNA is degraded, allowing the newly formed negative DNA strand to serve as a template for partial positive strand synthesis within the capsid [30] (Figure 2, steps 6 and 7).

The mature HBV capsid, now containing the rcDNA genome, can either shuttle back to the nucleus to replenish the cccDNA pool—a mechanism that supports HBV persistence—or move through the cytoplasm for assembly with the viral membrane. Viral assembly relies on interactions with multivesicular bodies, facilitating the encapsidated genome’s envelopment in a lipid bilayer embedded with HBsAg. This process results in the secretion of fully assembled progeny virions from the hepatocyte, ready to infect new cells and propagate the infection (Figure 2, steps 8–11).

Subviral particles self-assemble in the ER–Golgi intermediate compartment with lipids before being secreted via the constitutive secretory pathway. The HBsAg alone can efficiently produce highly immunogenic 20 nm HBsAg particles, the basis for most hepatitis B vaccines. Particle morphology depends on the S HBsAg-to-L HBsAg ratio, with low L HBsAg protein content favoring spherical particles and high L content inhibiting secretion and promoting filamentous forms. The L protein is essential for virus assembly and infectivity, while the M protein does not significantly affect particle morphology or formation (Figure 2, steps 8 and 9) [5].

This intricate series of steps—from NTCP-mediated entry and nuclear cccDNA formation to replication and assembly—illustrates HBV’s highly evolved mechanisms to establish and maintain chronic infection within hepatocytes. Understanding each step in this pathway offers critical insights into potential therapeutic targets aimed at interrupting the HBV life cycle.

## 4. The Formation and Dynamics of the cccDNA Reservoir

As outlined in the preceding paragraphs, the formation and replenishment of cccDNA are essential to the HBV life cycle and the persistence of chronic infection. Acting as the template for all viral RNAs, cccDNA ensures the continuous production of both the viral genome and associated proteins. To effectively target cccDNA for HBV eradication, a comprehensive understanding of the mechanisms and factors involved in its formation and maintenance is critical.

One notable feature of the cccDNA reservoir is the variability in copy numbers across individual cells depending on cellular conditions and the disease phase. In vitro studies using HepG2-NTCP cells, a common laboratory in vitro model for studying HBV infection, have estimated the cccDNA copy number per cell to be approximately 2.2 copies [15]. However, in patients with chronic hepatitis B, both the levels and activity of cccDNA fluctuate across different phases of the disease ranging from as low as 0.0032 to as high as 0.032 copies per cell. This variation can be attributed to factors such as immune and viral transcriptional activity, both of which fluctuate across different stages of HBV infection [18]. Additionally, monitoring viral RNA, particularly the 3.5 kb RNA transcript produced from cccDNA, provides insights into the transcriptional activity and potential replication capacity of the cccDNA reservoir in infected hepatocytes [31].

As shown in Figure 3, the conversion of HBV rcDNA to cccDNA begins soon after viral entry and proceeds through a multistep process. Research shows that this transformation occurs within the first 16 h post-infection, setting the stage for ongoing HBV replication early in the infection cycle [32]. Once established, cccDNA is remarkably stable, displaying longevity independent of frequent replenishment. Experimental evidence from HBcAg-mutant HBV constructs that have crippled nuclear import to replenish the cccDNA reservoir, for example, demonstrates that cccDNA levels remain constant (Figure 2, step 11), underscoring its self-sustaining nature in quiescent cells in vitro [33]. Even though cccDNA levels of quiescent primary human hepatocytes were shown to be stable in an HBV-infected humanized mouse model as well, in actively dividing hepatocytes, cccDNA levels can decrease significantly. This decline, observed both in vivo and in vitro, indicates that cccDNA lacks a nuclear retention mechanism for equal distribution during cell division, causing it to dilute over successive generations of dividing cells [34,35]. To counterbalance this dilution, HBV relies on the de novo synthesis of cccDNA through new infections and the reintegration of capsids into the nucleus, thus maintaining a sufficient reservoir of cccDNA in rapidly proliferating hepatocytes.

HBV cccDNA formation is a complex process involving host cellular enzymes and DNA repair machinery, as viral proteins play only limited roles in rcDNA repair. The biogenesis process begins with the removal of viral polymerase, which is covalently attached to the 5′ end of the rcDNA minus strand, blocking subsequent repair processes. Host enzymes, such as tyrosyl-DNA phosphodiesterases (TDPs) and FEN-1 endonucleases, along with proteases, play key roles in cleaving the polymerase and preparing rcDNA for further processing. This initial step yields a deproteinated rcDNA that serves as a precursor for cccDNA formation. To ensure the accuracy of the viral genome, the next step involves removing redundant terminal sequences (known as “r” sequences) from the rcDNA’s minus strand. Nucleases, including FEN-1, cleave these sequences, preventing potential errors during replication. Additionally, FEN-1 and possibly other nucleases help eliminate the 18-nucleotide-long RNA flap on the plus strand, which resembles an Okazaki fragment and impedes the synthesis of a complete viral genome. Following these preparatory steps, host DNA polymerases synthesize the missing plus strand and repair any gaps in the DNA. DNA ligases then seal the nicks on both strands, completing the conversion of rcDNA to the fully double-stranded cccDNA molecule (reviewed in detail in [28]).

Following rcDNA repair, cccDNA undergoes chromatinization to form a stable minichromosome. The host cell’s HIRA complex, composed of HIRA, ubinuclein-1 (UBN1), and CABIN1, facilitates this process by promoting histone deposition and nucleosome assembly [36]. Together with ASF1a, the HIRA complex deposits histone proteins onto the cccDNA, thus transforming the viral DNA into a chromatinized structure that can be transcriptionally regulated within the nuclear environment [36,37]. The minichromosome is further regulated by both viral and host proteins. Viral proteins, like HBcAg and polymerase, contribute to minichromosome formation but are not essential for cccDNA stability [36]. Host transcription machinery, specifically RNA polymerase II, controls the expression of cccDNA through four primary viral promoters (preS1, preS2, core, and X) and two enhancers (enhancer 1 and 2) [36]. Furthermore, cccDNA is subject to posttranscriptional modifications via methylation and acetylation, which modify chromatin states to support efficient viral transcription. Notably, these posttranscriptional modification patterns differ significantly between in vitro cell models and patient liver samples, suggesting that cccDNA transcriptional regulation adapts to distinct cellular environments [15]. In addition to chromatin modifications, cccDNA benefits from spatial organization within the nucleus. By localizing near regions rich in transcriptional activity, G-quadruplex structures, and liquid–liquid phase separation domains, cccDNA maximizes its transcription efficiency, supporting robust viral protein production and genome replication [38,39].

In summary, the persistence of HBV infection depends heavily on the stability and replenishment of the cccDNA reservoir. Once transcribed, viral proteins are synthesized at the ER and transported through the cell to support various stages of the HBV life cycle. The pgRNA transcribed from cccDNA acts as both a template for new viral genomes and encodes for essential viral proteins, like HBcAg and polymerase. Inside the viral capsid, the pgRNA is reverse-transcribed into rcDNA, allowing the mature nucleocapsid to either exit the cell to propagate infection or re-enter the nucleus to resupply the cccDNA pool (Figure 2). This cycle of cccDNA biogenesis and nuclear reimportation helps HBV to establish a long-term presence in the host liver, making it resilient to immune responses and current antiviral treatments.

## 5. Targeting cccDNA as a Therapeutic Approach Against Chronic HBV Infection

Complete eradication of chronic HBV infection has remained a formidable challenge due to the persistence of cccDNA in the nuclei of infected hepatocytes. Acting as a stable, self-renewing viral reservoir, cccDNA continuously produces all necessary viral RNAs and proteins, sustaining HBV replication and facilitating reactivation of the virus. Although current treatments effectively suppress viral replication, they do not eliminate cccDNA, making complete eradication of HBV challenging. Emerging therapeutic strategies are now focused on directly targeting cccDNA, potentially offering a complete cure by reducing or eliminating this reservoir and preventing HBV reactivation.

NAs, including entecavir and tenofovir, represent the primary treatment approach for chronic HBV. The primary goal of NAs is to stop viral replication, therefore decreasing viral load and preventing the circulating virus from infecting new hepatocytes and eventually stopping the related inflammatory response that may lead to liver damage and disease progression to hepatocellular carcinoma. By targeting HBV polymerase activity (Figure 3, steps 7), NAs prevent the conversion of pgRNA to rcDNA, thereby halting viral replication and limiting the formation of new viral particles. However, NAs only indirectly target the cccDNA reservoir, as they do not reduce cccDNA levels directly (Figure 3, steps 1 and 8). Earlier analogs such as lamivudine, adefovir, and telbivudine were phased out due to high rates of viral resistance after prolonged use [6]. However, with newer analogues such as tenofovir and entecavir, this appears no longer an issue. NAs function as prodrugs activated by cellular kinases into triphosphate forms, allowing them to compete with natural nucleotides and prematurely terminate HBV DNA synthesis. They act at critical stages during RNA- and DNA-dependent polymerization of rcDNA, often by inhibiting reverse transcription and protein priming of the polymerase (Figure 2 and Figure 3) [6,40]. Guanosine analogs can disrupt the initiation step of HBV protein priming, where dGTP is the initiating nucleotide, while adenosine analogs may interfere with the synthesis of dGAA trinucleotides. Entecavir triphosphate, for example, disrupts priming initiation, while tenofovir diphosphate interferes with later steps, including dAMP incorporation [40]. A study evaluating cccDNA levels before and after NAs treatment reported a 99.89% reduction in cccDNA levels, with 49% of patients having cccDNA levels below the detection limit [41]. Another study assessing pre- and post-treatment cccDNA levels in the livers of patients who underwent long-term NA treatment (3–8 years), followed by 24 weeks of pegylated IFN-α, revealed that only 13 out of 48 patients who achieved functional cure had undetectable cccDNA levels again using quantitiative PCR detection methods [42]. However, this outcome may depend on the sensitivity of the detection methods used in measuring low cccDNA levels, and improved detection technologies such as digital droplet PCR may update these data [31,43,44,45,46]. Notably, HBV has been shown to reactivate in patients after stopping NA therapy, highlighting the persistence of a small cccDNA reservoir even with long-term treatment [47].

Immunomodulation can be accomplished by pegylated IFN-α, but it also directly impacts cccDNA by recruiting host enzymes such as histone deacetylase 1 to the cccDNA minichromosome (Figure 3, step 5), potentially leading to transcriptional suppression. IFN-α may also reduce HBV RNA levels by destabilizing cccDNA, although the exact mechanisms remain unclear. Unlike NAs, IFN-α is thought to affect cccDNA transcriptional activity by reducing the number of activator post-translational modifications, rather than increasing the repressive post-translational mechanisms [48,49,50,51]. In addition to IFN-α, in vitro studies on interleukin-6 (IL-6) has shown potential in reducing cccDNA transcription by modulating posttranscriptional modification and altering transcription factor binding, suggesting that cytokine-mediated pathways could offer additional routes to suppress HBV replication [52]. However, while IFN-α and related cytokines can reduce cccDNA transcriptional activity, they are not capable of completely clearing the cccDNA reservoir.

Viral protein HBx can modify several cellular pathways and interact with various transcription factors. The discovery of the HBx protein recruitment on HBV cccDNA and its essential role in cccDNA transcriptional regulation has opened new therapeutic avenues. HBx was also shown to inhibit the host restriction factor Smc5/6 complex, which has identified HBx protein as a key player in enhancing viral gene transcription from cccDNA (Figure 3, step 5). Targeting HBx with small-interfering RNAs (siRNAs), with IFN-α in combination with siRNAs or small-molecule inhibitors, such as nitazoxanide and pevonedistat, have shown promise in cell line models and HBV-infected human liver chimeric mice [53,54,55,56,57,58]. Another natural compound sphondin has also been tested for its ability to inhibit HBx activity in vitro. A small molecule called ccc_R08 was reported recently to reduce HBV transcripts, proteins, and cccDNA levels in both HBV mouse models and humanized liver models, although further studies are required to assess its efficacy and off-target effects [59].

Emerging approaches are also exploring the blocking of HBV entry and nuclear import as methods to reduce rcDNA formation and, consequently, cccDNA replenishment (Figure 3, step 1). Bulevirtide, a synthetic lipopeptide currently in clinical trials for hepatitis D virus (HDV) treatment, functions as an HBV entry inhibitor by blocking the NTCP receptor on hepatocytes [60]. By preventing HBV entry, bulevirtide could limit the formation of new rcDNA, reducing the potential for cccDNA replenishment. Similarly, capsid assembly modulators have been developed to interfere with the HBV capsid’s ability to uncoat during de novo infection, thereby preventing nuclear reimport and subsequent rcDNA-to-cccDNA conversion ((Figure 3, steps 2, 7, and 8), [33,61]). However, both entry inhibitors and capsid assembly modulators primarily impact de novo infections and have minimal effect on pre-existing cccDNA levels.

Several small molecule inhibitors have been identified that can reduce cccDNA levels by disrupting various DNA repair factors essential for the rcDNA to cccDNA conversion (Figure 3, step 3). Experimental drugs that target DNA polymerases, FEN-1 endonucleases, ligases, and DNA checkpoint kinases (ATR and CHK1) have shown promise in in vitro cell lines and primary human hepatocyte models and biochemical assays. Because hepatocytes are largely non-dividing, selective inhibition of these pathways in liver cells may be possible without significant side effects. Strategies such as hepatocyte-targeted drug delivery and liver-specific activation of prodrugs are being investigated to increase the selectivity and safety of these approaches (reviewed in detail in [28]).

Cytokines like interferon-gamma (IFN-γ), tumor necrosis factor (TNF), and transforming growth factor-beta (TGF-β) have emerged as potential candidates for HBV therapy due to their ability to induce antiviral responses and target cccDNA indirectly. Studies have shown that activation of the lymphotoxin beta receptor (LTβR) or treatment with IFN-α can increase the expression of APOBEC3 enzymes, a family of cytidine deaminases with antiviral activity. APOBEC3 enzymes act by converting cytosines to uracils within DNA, which leads to mutations or DNA degradation (Figure 3, steps 3 and 4). These studies have suggested that the coordinated activity of APOBEC3A and APOBEC3B may degrade cccDNA, although the precise effectiveness and safety of such treatments remain to be fully elucidated [62,63,64]. Other studies also highlighted the potential of interferon-stimulated gene 20 (ISG20), a protein activated by IFN signaling, to target cccDNA in conjunction with APOBEC enzymes. Upon activation, ISG20 localizes to the nucleus and, in combination with APOBEC3, can target cccDNA for degradation in both hepatocytes and HBV-infected liver tissue [65]. IFN-α treatment was also documented to increase mRNA expression levels of not only APOBECs, but also other base excision repair pathway genes in patient HBV livers. This increase in the expression levels of base excision repair pathway genes reversely correlated with serum HBsAg levels and HBV viral loads. IFN-α responders had higher levels of APOBEC3A and APOBEC3B mRNA expression levels, but the effect of this increase on cccDNA levels was not studied [66]. Another study reported that baseline intrahepatic HBV DNA/cccDNA ratio and transcriptional activity of cccDNA can predict IFN-α treatment responses: a higher transcriptional activity makes intrahepatic cccDNA reservoir more susceptible for IFN-α targeting, and patients respond to IFN-α better [46]. While these findings are promising, only a subset of patients respond effectively to IFN-α, and prolonged use can cause significant side effects [67].

## 6. Concluding Remarks

While current therapies provide effective viral suppression, they fail to fully eradicate HBV due to the persistence of cccDNA. Novel approaches that target cccDNA, either by directly degrading it or by suppressing its transcriptional activity, offer the potential for a complete cure. As research continues, treatments that combine NAs; IFN-α; and novel small molecule inhibitors that target HBx, APOBEC, or ISG20 try to create a multifaceted approach to eliminate cccDNA and to reduce HBV reactivation risk after treatment cessation. Moreover, new drug delivery technologies and selective activation of antiviral pathways in the liver promise to enhance therapeutic precision, making HBV eradication a realistic goal. Further studies will be essential to refine these strategies, assess their long-term efficacy and safety, and better understand the pathways involved in cccDNA targeting to unlock effective cures for chronic HBV infection.

## Figures and Tables

**Figure 1 biomolecules-15-00062-f001:**
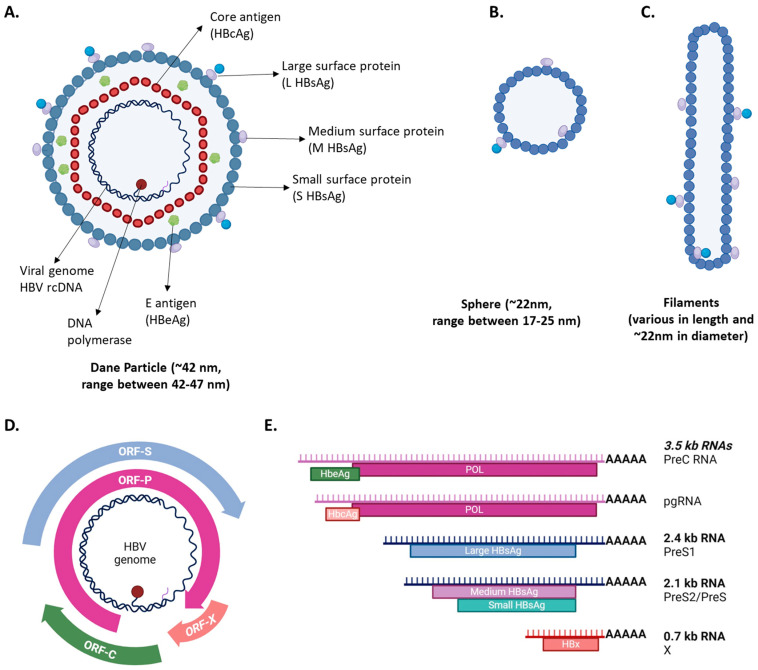
Schematic representation of (**A**) infectious HBV particles (Dane particles) and non-infectious (**B**) sphere and (**C**) filament subviral particles. Schematic HBV genome organization and open reading frames (ORFs) (**D**). (**E**) shows the HBV RNAs encoded from HBV cccDNA and the proteins (underneath the RNA molecules) produced from these RNAs. Common names of the RNAs and their lengths are shown on the right. Created in BioRender, Sari G., 2024 (BioRender.com).

**Figure 2 biomolecules-15-00062-f002:**
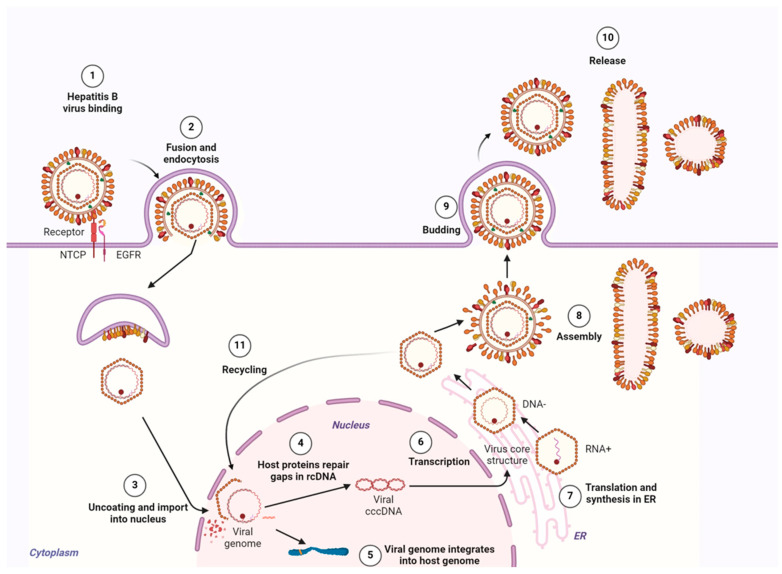
Schematic representation of HBV life cycle from viral entry into hepatocytes to viral secretion and nuclear reimport. For detailed information, please check Section 3. Created in BioRender (Adapted from “BioRender Disease Mechanisms—Infectious Diseases, Hepatitis B Virus Infection Cycle”, by BioRender.com).

**Figure 3 biomolecules-15-00062-f003:**
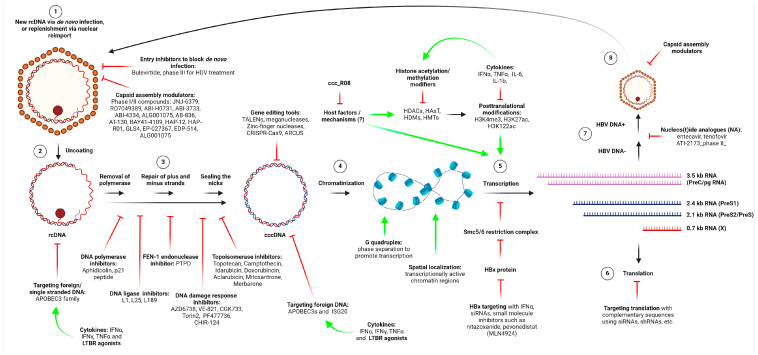
Schematic representation of the cccDNA minichromosome formation and viral transcription. HBV genomic DNA, rcDNA, conversion to viral minichromosome, cccDNA, and production of HBV transcripts is a multistep process, and almost every step of this process has been researched to be targeted as a novel anti-viral treatment strategy. Figure also summarizes the cccDNA targeting strategies. For detailed information, please check Section 4 and Section 5. Created in BioRender, Sari G., 2024 (BioRender.com).

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
