# Peer review of "HBV cccDNA: The Molecular Reservoir of Hepatitis B Persistence and Challenges to Achieve Viral Eradication"

_biomolecules, 2025, doi:10.3390/biom15010062_

Round 1

Reviewer 1 Report

Comments and Suggestions for Authors

The authors wrote an interesting contribution addressing  major biological issues associated with effective eradication of HBV infection, and specifically the removal  of  HBV cccDNA.

I  offer some suggestions to make this a little more lucid and focused for both the molecular biologist and the clinician (this reviewer belonging to the latter group)

My current understanding  re viral eradication:  We have an issues at 3 levels

1.     Humoral immunity: Circulating virus in the body  (and well controlled with antiviral, antibodies as needed,) Less mature immune system> increased risk carrier chronic infection etc.

2.     Cellular immunity: Infected hepatocytes killed by the cellular immune system  (if not by natural process, occasionally successfully done by using interferon therapy)

3.     Attempts to suppress viral replication. With current NAs this  can be done very effectively but    HBV cccDNA  is the remaining  big challenge and addressed  in this article. Whereas antivirals can suppress the virus, they do not destroy the intracellular viral replicative mechanism

I am completely missing element of host-response that the authors are very knowledgeable about.  Adds to risk of chronicity if system is immature, healthy adults will usually clear, immune -compromised may not. This adds also to the logic of what happens to neonates vs older people. What host response would help/not help to eradicate cccDNA?

The review would benefit from more briefly summarizing the non-cccDNA issues  (very well summarized in multiple Nature and other reviews)  because the reader gets a little lost in too many less focused elaborations and could benefit from a somewhat more crispy and focused summary of the current specific challenges

I think the introduction could be somewhat upgraded (see details)

Abstract   Chronic HBV infection is ....Current treatment is effective  in suppression of viral activity but fails to eradicate the virus

Introduction  

Highly contagious if one is exposed  to infected body fluids   (but we may kiss, we can breastfeed, labeling highly contagious in general  adds to stigmatization)

Sexual transmission is left out

Many cases occur without jaundice and are therefore not recognized as hepatitis

NAs: Life-long tends to be favored, because although many people will do fine after having viral suppression for a number of years, a significant % may have very serious  and risky flare-ups. (ref)

The sentence re costs come a little out of the blue.  Put in a context: 

Eradication encounters many biomedical challenges  as will be outlined, but is also costly and includes  the struggles to get all newborns vaccinated according to the WHO guidelines and the costly maintenance therapy for chronically infected patients (ref).

Parts 2-4 . HBV genome ..... Are all details needed for the cccDNA issue including Dane particle etc. or is more focused selection helpful

Intractable challenge   should it be  ‘as yet unsurmountable’?

Part 5 This is wordy and could benefit from editing. Help the reader Suggest table with clear dissection of agent specific target etc.

Would  more precisely compare /define  specific goals of therapy/therapeutic  challenge

              NA   reduce viral load and prevent.....

Then suddenly Pegylated interferons known for their immunomodulary effects:  Immunomodulation may be accomplished by......

             Reorganizing  would help

Author Response

Reviewer #1 – Comment #1: ‘I am completely missing element of host-response that the authors are very knowledgeable about.  Adds to risk of chronicity if system is immature, healthy adults will usually clear, immune -compromised may not. This adds also to the logic of what happens to neonates vs older people. What host response would help/not help to eradicate cccDNA?’ The primary focus of our current review is on the molecular and biochemical aspects of cccDNA biology, which already very complicated and requires extensive explanation. In this context, the chronicity is already achieved and the cccDNA reservoir is already formed, host factors and immune system have been already hijacked. Given the breadth of the molecular and biochemical aspects of cccDNA field, a detailed discussion of the host response is beyond the scope of this review. We aim to maintain a focused narrative on cccDNA-specific mechanisms. That said, we will ensure that the manuscript explicitly clarifies this scope and will reference relevant literature on host-response factors for readers interested in those aspects.

Reviewer #1 – Comment #2: The review would benefit from more briefly summarizing the non-cccDNA issues  (very well summarized in multiple Nature and other reviews)  because the reader gets a little lost in too many less focused elaborations and could benefit from a somewhat more crispy and focused summary of the current specific challenges.’ As stated in the previous comment, due to the wide scope of the molecular and biochemical aspects of cccDNA, this review focuses on the detailed molecular and biochemical features of cccDNA biology, as well as the current strategies available to target it. Furthermore, as you pointed out, there have been excellent review articles published recently:

https://doi.org/10.1038/s41575-022-00724-5

https://doi.org/10.1038/s41575-024-00946-9

https://doi.org/10.1016/j.coi.2022.102207

Reviewer #1 – Comment #3: ’I think the introduction could be somewhat upgraded (see details)

Abstract   Chronic HBV infection is ....Current treatment to eradicate the virus’ is effective  in suppression of viral activity but fails’

Introduction  

Highly contagious if one is exposed  to infected body fluids   (but we may kiss, we can breastfeed, labeling highly contagious in general  adds to stigmatization)

Sexual transmission is left out’

Many cases occur without jaundice and are therefore not recognized as hepatitis We agree with the reviewer and the jaundice is only mentioned in the second paragraph for the symptomatic cases.

NAs: Life-long tends to be favored, because although many people will do fine after having viral suppression for a number of years, a significant % may have very serious  and risky flare-ups. (ref)’.

The sentence re costs come a little out of the blue.  Put in a context.

Response to Comment#3: Dear reviewer, we updated the abstract and the introduction accordingly. Wording has been updated as well such as’ HBV is an infectious virus’ instead of highly contagious, and ‘transmission can also occur through contact with infected blood through shared syringes and contaminated medical equipment and sexual contact with an infected person. Additional references are included (ref.9-11) regarding NA treatment and flares upon NUC stop studies. Please have a look at the Abstract and Section 1. Introduction for further details.

Reviewer #1 – Comment #4: ‘Parts 2-4 . HBV genome ..... Are all details needed for the cccDNA issue including Dane particle etc. or is more focused selection helpful’ Because viral replication and replenishment of the cccDNA can be targeted at several different steps, directly or indirectly, we believe that the provided information paves the path to be able to better understand the later sections on treatment strategies.

Reviewer #1 – Comment #5: Intractable challenge   should it be  ‘as yet unsurmountable’?’ The sentence is updated as: Complete eradication of chronic HBV infection has remained a formidable challenge due to the persistence of cccDNA in the nuclei of infected hepatocytes.

Reviewer #1 – Comment #6: Part 5 This is wordy and could benefit from editing. Help the reader Suggest table with clear dissection of agent specific target etc.’ We now numbered the steps in the cccDNA biogenesis pathway Figure (Fig.3) and ref.’d the steps in Part 5 to help the reader.

Reviewer #1 – Comment #7: ‘Would  more precisely compare /define  specific goals of therapy/therapeutic  challenge…..NA   reduce viral load and prevent.....Then suddenly Pegylated interferons known for their immunomodulary effects:  Immunomodulation may be accomplished by......Reorganizing  would help’. Part5 is updated accordingly and we reworded the information regarding different treatments. References to Fig.3 are added as well. Please see pages 9 and 10 for further details.

Reviewer 2 Report

Comments and Suggestions for Authors

This manuscript reviews how the Hepatitis B virus (HBV) develops in humans, the synthesis of covalently closed circular DNA (cccDNA) and its role in guiding HBV development, the challenges cccDNA poses to curative treatments, and the ongoing efforts to target cccDNA. The review is well-structured, transitioning smoothly from general processes to detailed discussions. The figures are organized and informative. However, some minor adjustments are recommended:

1. Expand on the description of cccDNA formation and ongoing curative treatment efforts targeting cccDNA. Currently, the abstract generalizes these topics in one sentence. Providing more details will improve clarity and reader engagement.

2. In figure1,

·      Use standardized abbreviations for surface proteins: large surface protein (L HBsAg), medium surface protein (M HBsAg), and small surface protein (S HBsAg).

·      The spherical subviral particles are composed of 90% M HBsAg and 10% S HBsAg, with no L HBsAg present. Remove the L HBsAg from the depiction of spherical subviral particles in Figure 1B.

·      Add the particle diameters to Figure 1A, B, and C for clarity. Dane particles: 42 nm; spherical SVPs: 25 nm; filamentous SVPs: 22 nm

3. Include a paragraph on the mechanisms by which subviral particles are formed and exported from host cells. This addition will provide a more comprehensive understanding of the HBV life cycle.

4. The figures contain many details, but the references to them in the text are unclear. For example, in Section 5, there are frequent references to Figure 3, but the specific parts of the figure being discussed are not indicated.

5. Remove the title of Figure 3 (“Figure-3” in Figure 3)

Author Response

Reviewer #2 – Comment #1: ‘Expand on the description of cccDNA formation and ongoing curative treatment efforts targeting cccDNA. Currently, the abstract generalizes these topics in one sentence. Providing more details will improve clarity and reader engagement.’ Abstract is expanded accordingly.

Reviewer #2 – Comment #2: In figure1, Use standardized abbreviations for surface proteins: large surface protein (L HBsAg), medium surface protein (M HBsAg), and small surface protein (S HBsAg).’ We have updated the acronyms in Fig.1A.

‘The spherical subviral particles are composed of 90% M HBsAg and 10% S HBsAg, with no L HBsAg present. Remove the L HBsAg from the depiction of spherical subviral particles in Figure 1B.’ I think this is a typo because spherical subviral particles are composed of 90% S HBsAg and 10% M HBsAg and with trace amounts of L HBsAg (PMID: 39264996). We agree with reviewer #1 that L-HBsAg proportion of subviral particles is typically very low because it is primarily required for envelope formation in infectious particles and its overexpression can inhibit SVP formation (PMID: 17206755, PMID: 39264996, PMID: 18524834, PMID: 30391399). That’s why spherical subviral particle is depicted as containing only 1 L HBsAg.

‘Add the particle diameters to Figure 1A, B, and C for clarity. Dane particles: 42 nm; spherical SVPs: 25 nm; filamentous SVPs: 22 nm’. Corresponding diameters are included.

Reviewer #2 – Comment #3: Include a paragraph on the mechanisms by which subviral particles are formed and exported from host cells. This addition will provide a more comprehensive understanding of the HBV life cycle.’ A paragraph about subviral particles is added in Part-3. Please check page 6 for further details.

Reviewer #2 – Comment #4: ‘in Section 5, there are frequent references to Figure 3, but the specific parts of the figure being discussed are not indicated.’ Section-5 and Fig.3 are updated accordingly.

Reviewer #2 – Comment #5: ‘Remove the title of Figure 3 (“Figure-3” in Figure 3)’. Removed.

Round 2

Reviewer 1 Report

Comments and Suggestions for Authors

Thank you for updating various aspects.  I made a few minor sticky note comments that you are welcome to use

With so many abbreviations in your (beautiful) figures, I would reflect for a moment if abbreviations are all covered in text. Example EFGR 

Author Response

Reviewer #1 – Comment #1: ‘Thank you for updating various aspects. I made a few minor sticky note comments that you are welcome to use’. Using the sticky notes, we have updated the text/wording. All changes are highlighted in red.

Reviewer #1 – Comment #2: With so many abbreviations in your (beautiful) figures, I would reflect for a moment if abbreviations are all covered in text. Example EFGR.’ All abbreviations including EGFR (not EFGR) are covered either in the text or in the Appendix A section. For EGFR, please have a quick look at page 4, lines 129 and 130.